# Dosimetric Comparison and Selection Criteria of Intensity-Modulated Proton Therapy and Intensity-Modulated Radiation Therapy for Adaptive Re-Plan in T3-4 Nasopharynx Cancer Patients

**DOI:** 10.3390/cancers16193402

**Published:** 2024-10-05

**Authors:** Mincheol Ko, Kyungmi Yang, Yong Chan Ahn, Sang Gyu Ju, Dongryul Oh, Yeong-bi Kim, Dong Yeol Kwon, Seyjoon Park, Kisung Lee

**Affiliations:** 1Department of Radiation Oncology, Samsung Medical Center, School of Medicine, Sungkyunkwan University, Seoul 06351, Republic of Korea; mincheol.ko@samsung.com (M.K.); kyungmi.yang@samsung.com (K.Y.); dongryul.oh@samsung.com (D.O.); dy82.kwon@samsung.com (D.Y.K.); seyjoon@yuhs.ac (S.P.); 2Department of Bio-Convergence Engineering, Korea University, Seoul 02841, Republic of Korea; kisung@korea.ac.kr; 3Department of Digital Health, Samsung Advanced Institute for Health Sciences & Technology, Sungkyunkwan University, Seoul 06355, Republic of Korea; dudql321@naver.com

**Keywords:** nasopharyngeal cancer, proton therapy, intensity-modulated radiotherapy, helical tomotherapy, organs at risk, radiation injury, radiotherapy treatment planning

## Abstract

**Simple Summary:**

Nasopharynx cancer treatment often involves radiation therapy, but choosing the best method can be challenging. This study compares two advanced radiation techniques: one using X-rays and another using protons. Both aim to target the tumor while sparing healthy tissues, but their effectiveness can vary based on the cancer’s location and stage. Researchers want to determine which technique works best for different patients, particularly those with cancer close to sensitive brain structures. By analyzing treatment plans for 28 patients, we developed guidelines to help doctors choose the most suitable technique. This research could lead to more personalized and effective treatments, potentially improving outcomes and quality of life for patients with nasopharynx cancer. The findings may also help healthcare providers use resources more efficiently, benefiting both patients and the broader medical community.

**Abstract:**

Background: Proton therapy requires caution when treating patients with targets near neural structures. Intuitive and quantitative guidelines are needed to support decision-making concerning the treatment modality. This study compared dosimetric profiles of intensity-modulated proton therapy (IMPT) and intensity-modulated radiation therapy (IMRT) using helical tomotherapy (HT) for adaptive re-planning in cT3-4 nasopharyngeal cancer (NPCa) patients, aiming to establish criteria for selecting appropriate treatment modalities. Methods: HT and IMPT plans were generated for 28 cT3-4 NPCa patients undergoing definitive radiotherapy. Dosimetric comparisons were performed for target coverage and high-priority organs at risk (OARs). The correlation between dosimetric parameters and RT modality selection was analyzed with the target OAR distances. Results: Target coverages were similar, while IMPT achieved better dose spillage. HT was more favorable for brainstem D_1_, optic chiasm D_max_, optic nerves D_max_, and p-cord D_1_. IMPT showed advantages for oral cavity D_mean_. Actually, 14 IMPT and 14 HT plans were selected as adaptive plans, with IMPT allocated to most cT3 patients (92.9% vs. 42.9%, *p* = 0.013). The shortest distances from the target to neural structures were negatively correlated with OAR doses. Receiver operating characteristic curve analyses were carried out to discover the optimal cut-off values of the shortest distances between the target and the OARs (temporal lobes and brainstem), which were 0.75 cm (AUC = 0.908, specificity = 1.00) and 0.85 cm (AUC = 0.857, specificity = 0.929), respectively. Conclusions: NPCa patients with cT4 tumor or with the shortest distance between the target and critical neural structures < 0.8 cm were suboptimal candidates for IMPT adaptive re-planning. These criteria may improve resource utilization and clinical outcomes.

## 1. Introduction

Nasopharynx cancer (NPCa) is a rather rare disease, with a global age-adjusted incidence rate of 1.5 per 100,000 person-year in 2020 [1]. The incidence rate of NPCa has been reported to be the highest in eastern Asia (including China) and south-eastern Asia. High-dose radiation therapy (RT), frequently assisted with systemic chemotherapy, has long played a pivotal role in treating NPCa patients [2,3,4]. Unlike other head and neck cancer types, NPCa sometimes involves the skull base (cT3 disease) or invades the cranial cavity (cT4 disease) by virtue of its anatomic proximity. Recent advances in photon-based RT techniques, typically including highly conformal radiation delivery through intensity modulation and image guidance, have contributed to improved local control and survival outcomes with less severe toxicity profiles [5]. Despite these benefits, high-dose RT is often associated with both early and delayed radiation toxicities, especially when the target is located very close to the surrounding normal organs at risk (OARs).

This risk level becomes inevitably high in treating nasopharynx cancer (NPCa) patients with advanced cT3-4 lesions, where the target is very close to neural structures including the temporal brain, brainstem, and several cranial nerves including optic nerves [6]. By virtue of Bragg–Peak phenomenon, proton beam therapy (PBT) has been regarded an alternative to photon-based RT techniques in treating patients with skull base tumors including chordoma and chondrosarcoma, with the advantage of less severe neural tissue damage [7,8,9]. Incorporation of PBT, either as a sole modality or in combination with photon-based RT, is naturally assumed to be more advantageous over photon therapy alone by reducing the neural tissue damage risk in treating cT3-4 NPCa patients. Alterio et al. reported comparable toxicity and efficacy following either the mixed-beam approach of serial intensity-modulated radiotherapy (IMRT) and PBT or IMRT alone in this setting [10].

Researchers have applied a few strategies to increase the therapeutic ratio in treating patients with oropharynx cancer and NPCa, which have evolved over time [11,12]. The most recent one could be represented as the serial combination of IMRT and intensity-modulated proton therapy (IMPT) including a simultaneous integrated boost (SIB), moderately hypofractionated dose schedule, and adaptive re-plan; this has been used since 2019, when the clinical application of PBT became available in our institute. We have generated rival plans for an adaptive re-plan, one by IMRT based on helical tomotherapy (HT, TomoHD, Accuray, Sunnyvale, CA) and the other by IMPT, in treating cT3-4 NPCa patients. After comparing the dosimetric parameters of the paired rival plans, especially focusing on high-risk OARs including neural structures, we determined the actual RT modality for the adaptive re-plan. Our effort has been based on the recent knowledge that the relative biological effectiveness (RBE) of a proton beam, based on long-term outcomes, is not 1.1 but is 1.18 instead for brain tissue [13]. Therefore, we realized that PBT should be performed more cautiously in treating patients with locally advanced NPCa, in whom the targets are very close to neural OARs. 

In treating locally advanced NPCa patients, there are many occasions when it is very hard to satisfy all the OAR constraints, and the radiation oncologist in charge should carefully consider important decision-making points before selecting the appropriate RT modality. Recently, Vai et al. suggested a decision-support tool to select IMPT based on the OAR mean dose and normal tissue complication probability (NTCP) models in treating NPCa patients [14]. Another study, which used the NTCP model for the temporal necrosis risk in skull base tumors and head and neck cancers, suggested a few risk factors including old age, high prescription dose, high D1cc, and hypertension, respectively [15]. Despite their good prediction performance, this model was hardly considerable for NPCa patients because the main risky neural structures, such as the brainstem or optic tracks, were not fully considered. IMPT has been recently developed and applied in treating NPCa patients. Therefore, the pre-existing NTCP models may not seem to adequately evaluate the neural structure’s potential damage, which usually has a relatively lower incidence and longer latency compared to other organs. In fact, Zheng et al. analyzed the late toxicities in long-term NPCa survivors treated with IMRT, in which the incidences of neural injuries were around 10% in cT3-4 patients [16]. Considering that the RBE of brain tissue in PBT is 1.18 [13], neural toxicity after PBT could be even higher. Also, once the neural toxicity occurs, it is usually difficult for the victims to achieve full recovery, and they usually have to suffer from long-term impaired quality of life. From this perspective, to support wise decision-making between IMRT and IMPT, intuitive and quantitative dosimetric guidelines are very useful in clinical practice.

This study intended to compare and evaluate the dosimetric profiles of paired rival plans generated for adaptive re-plans focusing on high-risk OARs, including the neural structures, in treating cT3-4 NPCa patients. Additionally, we aimed to establish criteria for selective rival plans.

## 2. Materials and Methods

### 2.1. Patients

From April 2019 to August 2020, 104 NPCa patients underwent definitive RT at the authors’ institute. Among them, we selected 28 consecutive cT3-4 NPCa patients, staged according to the American Joint Committee on Cancer 8th edition [17] through a thorough retrospective medical record review.

This study complied with the Declaration of Helsinki and was approved by the Institutional Review Board (IRB) of the Samsung Medical Center (No. 2021-05-046). The requirement for written informed consent was waived by the IRB due to the retrospective nature of this study.

### 2.2. Radiation Therapy

All patients underwent computed tomography (CT)-based simulation in the supine position with a customized thermoplastic mask (Aquaplast RT, Q-fix, Avondale, PA, USA). On simulation CT images, the gross tumor volume (GTV) was delineated based on all available clinical information including physical examination and diagnostic images such as CT, positron emission tomography–CT and/or magnetic resonance imaging. The clinical target volume (CTV) was delineated by adding 3–5 mm margins in all directions from the GTV of the primary tumor and typically included the adjacent structures based on the authors’ policy and the international guidelines [11,18]. The planning target volume (PTV) was generated with a 3 mm isotropic expansion from the GTV (P-GTV) and CTV (P-CTV), respectively. Delineation of the OARs, including the spinal cord, brainstem, temporal lobes, optic nerves, optic chiasm, lens, eyeballs, cochlea, extended oral cavity, constrictor muscles, parotid glands, submandibular glands (SMGs), and thyroid gland, was carried out according to the consensus guidelines [19,20]. The p-cord was delineated by expansion of a 3 mm margin from the spinal cord. The planned dose schedule was to deliver 67.2 Gy to P-GTV and 56 Gy to P-CTV in 28 fractions by the SIB concept.

### 2.3. Rival Plans for Adaptive Re-Plan

Initially, 16 fractions were delivered by HT to deliver 38.4 Gy to GTV and 32 Gy to CTV, followed by 12 fractions of adaptive re-plan either by HT or IMPT (Sumitomo Heavy Industries, Ltd., Tokyo, Japan) to deliver 28.8 Gy to GTV and 24 Gy to CTV, respectively. Regardless of the RT modality during the adaptive re-plan, the dose prescription policies were the same (minimum 95% to the PTV, while keeping the maximum dose within 107% of the prescription dose). RT plan generation was carried out by Accuray Precision^®^ (version 1.1.1.1, Accuray, Sunnyvale) for HT plans and RayStation^®^ (version 8.1, RaySearch Laboratories, Stockholm, Sweden) for IMPT plans, respectively (Appendix A).

For the HT plan, the plan conditions included a field width of 2.5 cm, modulation factor of 2.0, and pitch of 0.287. The dynamic jaw mode (TomoEDG-ETM, Accuray) was employed to improve the longitudinal dose conformity [21]. The final dose calculation was conducted using the collapsed-cone convolution algorithm. For IMPT, all the doses were defined in RBE-weighted grays (Gy(RBE)) assuming an RBE of 1.1 and single-field optimization (SFO) based on active line scanning with a range shifter (4 cm water equivalent thickness). The typical beam arrangements were three beams (one posterior and bilateral anterior oblique ports) or two beams (one posterior and ipsilateral anterior oblique) (Appendix A). These beam arrangements were chosen to optimize the target dose coverage while minimizing the uncertainties along the beam paths. Proton beam energies of 70 MeV (8.4 mm spot size, inline, in the air) to 180 MeV (4.7 mm spot size, inline, in the air) were used. Robust optimization considered ±3 mm setup errors and ±3.5% range uncertainties (21 scenarios) [22,23]. The Monte-Carlo dose calculation algorithm was used for dose calculation with 1% statistic uncertainty.

The actual RT modality for adaptive RT was chosen based on the comparative evaluation of the rival plans generated by HT and IMPT, considering the target coverage and normal organ constraints based on Quantitative Analyses of Normal Tissue Effects in the Clinic (QUANTEC) [24]. Finally, 14 IMPT plans and 14 HT plans were chosen as the adaptive plan, respectively. 

### 2.4. Dosimetric Parameters

To assess the competing plan quality, we selected the following target-related dosimetric parameters: the minimum doses received by 98% (D98), 50% (D50), and 2% (D2); homogeneity index (HI) [25]; and conformity index (CI) [26] for the GTV and CTV. To evaluate the dose fall-off gradient near the targets, we measured the dose spillage, defined as (VX% outside of the target volume)/target volume, where VX% is the volume covered by the X% isodose surface of the prescription dose [27,28]. The dose spillages were categorized as high- (X = 90), intermediate- (X = 50), and low-dose spillage volume (X = 25), and we compared the high-dose spillage for the GTV and CTV and the intermediate- and low-dose spillages for the CTV. The OAR-related dosimetric parameters included D1 (Gy) of the p-cord, brainstem, temporal lobes, and Dmax (Gy) of the optic chiasm and optic nerves, Dmax (Gy) of the lens, D1 (Gy) of the oral cavity, and Dmean (Gy) of the oral cavity, eyeballs, cochlea, constrictor muscles, parotid glands, submandibular glands, and thyroid gland. D1, Dmax, and Dmean were defined as the doses received by 1% of the volume, maximum, and mean dose to the corresponding OARs.

### 2.5. Shortest Distance between the Target and OARs

To evaluate the impact of the distance between the targets and high-priority OARs, we calculated the three-dimensional shortest distance between the target and each corresponding OAR (target to OAR distance) using the in-house developed software, which was based on the algorithm of Euclidean distance transformation [29]. 

### 2.6. Statistical Analysis

The dosimetric parameters of the rival plans were analyzed and compared using a paired *t*-test, while a *t*-test was used to compare the independent variables. Also, a correlation analysis, linear regression, and receiver operating characteristic (ROC) curve analysis were used to evaluate the relationship between the target to OAR distance and OAR doses. The ROC curves where decisions made in favor of IMPT were considered ‘positive’ and those made for HT were considered ‘negative’ were drawn with the GTV or CTV to OAR distances to calculate the area under the curve (AUC) and the cut-off values to select IMPT as the adaptive plan; the sensitivity refers to the probability of choosing IMPT for patients who are favorable for IMPT, while the specificity refers to the probability of choosing HT over IMPT for patients who are unfavorable for IMPT. All statistics and visualization of the graphs in this study used R 4.0.3 (R Development Core Team, Vienna, Austria).

## 3. Results

### 3.1. Patient Characteristics

The characteristics of the patients are summarized in Table 1. The Eastern Cooperative Oncology Group (ECOG) performance status of all patients was from 0 to 1, the median age was 50 years, and three-quarters were male. The clinical tumor stages were cT3 in 19 patients (67.9%) and cT4 in 9 (32.1%). Twenty-four patients (85.7%) underwent upfront definitive RT with concurrent chemotherapy (CCRT), where four (14.3%) received induction chemotherapy before CCRT. The median shortest distance from the GTV or CTV to the brainstem and temporal lobes was generally shorter (0~0.7 cm) than that to other OARs (1.1~2.15 cm).

### 3.2. Dosimetric Comparison

The paired scatter plots of the OAR doses by HT and IMPT plans are shown in Figure 1. Overall, more favorable dosimetric profiles were achieved by the HT plan with respect to the high-priority OARs including D1 of the brainstem (Figure 1A), Dmax of the optic chiasm (Figure 1C), Dmax of the optic nerves (Figure 1D), and D1 of the p-cord (Figure 1E), respectively. On the contrary, superior dosimetric profiles were achieved by the IMPT plan in terms of the Dmean of the oral cavity (Figure 1J). Similar dosimetric profiles, however, were achieved in other OARs by either plan. It is interesting to note that the range of scatter distribution of the Dmax of the optic chiasm (Figure 1C) and Dmax of the optic nerves (Figure 1D) was widely distributed when compared with other OARs. 

Table 2 summarized the dosimetric parameters generated by two RT techniques. The IMPT plan demonstrated a superior CI of GTV (HT 0.34 ± 0.08 vs. IMPT 0.40 ± 0.09, *p* < 0.001), and the HT plan exhibited a better HI of GTV (HT 1.03 ± 0.01 vs. IMPT 1.04 ± 0.01, *p* < 0.001); however, no significant difference was apparent in the HI and CI of CTV. Likewise, more favorable profiles were achieved by the IMPT plan with high-dose spillage of GTV (5.00 ± 2.49 vs. 3.87 ± 1.74, *p* < 0.001) and low-dose spillage of CTV (21.18 ± 2.92 vs. 15.96 ± 2.52, *p* < 0.001). Among the high-priority OARs, the HT plan generated more favorable profiles over the IMPT plan with respect to D1 of the brainstem (16.32 ± 3.78 vs. 20.52 ± 3.74, *p* < 0.001), Dmax of the optic chiasm (7.36 ± 7.24 vs. 14.07 ± 7.76, *p* < 0.001), Dmax of the optic nerves (11.33 ± 8.66 vs. 17.77 ± 7.47, *p* < 0.001), and D1 of the p-cord (10.37 ± 2.42 vs. 13.66 ± 4.70, *p* < 0.001), respectively. Meanwhile, no significant difference was shown for D1 of the temporal lobes. Among the low-priority OARs, the HT plan was more favorable for D1 of the oral cavity, Dmean of the parotid glands, Dmean of the SMGs, and Dmean of the thyroid, while the IMPT plan was advantageous for Dmax of the lens, Dmean of the eyeballs, Dmean of the cochlea, Dmean of the oral cavity, and Dmean of the constrictor muscles, respectively. 

The exposed doses to the high-priority OARs were compared according to cT stages (Figure 2). Regardless of cT stages, most OAR doses were consistently lower in the HT plans than in the IMPT, while the temporal lobes were highly exposed, though not significantly, to both treatment modalities.

### 3.3. Target to OAR Distance

A more favorable dose distribution was achieved in most OARs by the HT plans than by the IMPT plans, except when the target to p-cord distance was over 2~2.5 cm (Figure 3 and Appendix A). Overall, the shortest target to OAR distances and OAR doses were negatively correlated, as predicted. In detail, the brainstem and temporal lobes had a relatively stronger correlation and higher R^2^ of linear regression with the GTV to OAR distance than other OARs, while the optic chiasm and optic nerves showed a higher correlation with the CTV to OAR distances than the brainstem and temporal lobes. The absolute values of regression coefficient, which reflected the degree of dose decrease per 1 cm distance, were generally similar or higher in the HT plans. In the p-cord, however, this correlation was relatively weaker than other OARs in the HT plans.

### 3.4. Comparison of the Selected Adaptive Plans

The actual RT modality for the adaptive re-plan was selected by the radiation oncologists in charge after a comparative evaluation of both the HT and IMPT plans, and the patients’ characteristics and the target to OAR distances of 14 IMPT and 14 HT plans are detailed in Table 1 and Table 2. Most characteristics were not different except cT stages (*p* = 0.013), where the IMPT plan was most commonly allocated to patients with cT3 tumors (13/14, 92.9% in IMPT vs. 6/14, 42.9% in HT). For patients with cT4 tumors, HT was predominantly selected (8/9, 88.9% in HT). Also, the shortest distance between the GTV or CTV and the high-priority OARs was significantly longer in IMPT patients than in HT patients.

The ROC curves to select IMPT as the adaptive plan are shown in Figure 4. The values of the AUC with the GTV to OAR distances were the highest in the order of the temporal lobes, brainstem, optic chiasm, optic nerves, and p-cord (AUC = 0.908, 0.857, 0.842, 0.770, and 0.686, respectively), and the cut-off values for IMPT were 0.75 (sensitivity 0.714 and specificity 1.000), 0.85 (sensitivity 0.643 and specificity 0.929), 1.95, 2.00, and 1.30 cm, respectively. However, with the CTV to OAR distances, values of AUC were higher in the optic chiasm and optic nerves (0.920 and 0.916) than the brainstem and temporal lobes (0.821 and 0.750). 

## 4. Discussion

There has been a myth that PBT may be superior to photon-based RT techniques in generating more favorable dose distribution around the critical OARs by virtue of the Bragg–Peak phenomenon [15,30]. In treating nasopharyngeal cancer (NPCa) patients, based on our clinical experience, the physical advantage of the Bragg–Peak phenomenon was usually effective in all NPCa patients, typically leading to a reduced oral cavity dose and less severe pain. This advantageous feature of IMPT, however, was not always effective with respect to the critical neural structures, which were usually closely located to the target volumes. We intended to report a plan comparison study focused on cT3-4 NPCa patients whose targets are close to the critical neural OARs. 

There have been only limited number of studies that directly compared PBT against IMRT in treating locally advanced NPCa patients. Mintatogawa et al. reported a plan comparison study between adaptive IMRT and IMPT in ten cT1-3 NPCa patients, which had similarity to ours [30]. Alterio et al. also reported that the mixed-beam approach with IMPT showed less oral cavity dose and less severe acute mucositis in cT3-4 NPCa patients [31]. However, the limited follow-up period for the IMPT patients in these studies poses challenges in observing late complications, including neural toxicities. The determination of which patient groups might derive the most benefit from the limited resources of PBT remains an open question, and it is hard to make a reasonable clinical decision among the NPCa patients. 

In our institute, IMPT plans have been compared with rival plans by HT before determining the actual RT modality for an adaptive re-plan for NPCa patients. All the paired rival plans were generated along the same and consistent policy of simulation, contouring, and dose schedules on each patient. Though the proton beam’s Bragg–Peak phenomenon has been regarded as the most advantageous physical feature over the photon beam [32], in clinical practice, we have frequently encountered situations in which IMPT plans were not more favorable than HT plans, especially in treating cT3-4 NPCa patients. Paradoxically, some IMPT plans may exhibit worse dose fall-off profile than HT plans’ dose fall-off profiles near critical structures like the brainstem or optic tracts. In this context, we compared the paired IMPT and HT plans generated for the adaptive re-plan on 28 consecutive cT3-4 NPCa patients who were treated at the authors’ institute. As shown in the scatter plots of Figure 1, the exposure doses at the neural structures close to the target were lower in HT than in IMPT plans. The scatter plots and difference in the mean OAR doses (Table 2) were well correlated with our real-world experience, as mentioned above. As expected, the obvious difference between the HT and IMPT plans was evident in the low-dose spillage outside the target area (21.18 vs. 15.96, *p* < 0.001). Lesser low-dose spillage was also found in the mean dose of the oral cavity, and the relevant clinical benefit of IMPT was confirmed as less frequent and less severe acute oral mucositis, which was reported by us previously [33]. In the current study, the radiation oncologist determined the actual adaptive re-plan modality mainly based on the target coverage and dose constraints of high- and low-priority OARs, after objective and subjective comparison of the paired rival plans. Finally, the actual selected techniques were HT in 14 patients (50.0%) and IMPT in 14 patients (50.0%). Interestingly enough, two patient groups showed different clinical characteristics: the cT4 stage was more frequently observed in the HT group than in the IMPT group (57.1% versus 7.1%, *p* = 0.013). This differential selection in this study was well-matched with the results of direct comparison between HT and IMPT. All dosimetric parameters of the high-priority OARs, except that of the temporal lobes, were significantly unfavorable in the IMPT plans, even in the cT3 and cT4 patients (Figure 2 and Table 2). 

Apart from the cT stage, the optimal distance from the target to OARs for selecting an IMPT plan was uncertain. We hereby propose a novel concept of a three-dimensional calculation of the shortest distances between the targets and OARs as quantitative guidance. As expected, the shortest distances and OAR doses showed a strong negative correlation in both techniques (Figure 3). However, contrary to this inverse correlation hypothesis, HT was superior to IMPT on the condition that the shortest distances were within 2~3 cm in most OARs except for the p-cord. Nevertheless, as mentioned above, there were a few more conditions that need to be considered before selecting the modality, and we would propose the shortest distance to the critical OARs. In the ROC curve analysis, the cut-off values of the distance that showed high specificity to the brainstem and temporal lobes were 0.85 cm and 0.75 cm, respectively. This indicates that IMPT would be less selected as the treatment modality if the shortest distance from the target to the brainstem or temporal lobes was less than these cut-off values. Similarly, the shortest cut-off distance in case of the optic chiasm and optic nerves was about 1.5 cm, as these structures were located less close to the target. Supporting our results, the rapid dose fall-off between the target and OAR was closely related to the lateral penumbra width. The lateral penumbra width of the proton beam is usually narrower than the photon beam before the depth of Bragg–Peak, which, however, broadens with an increasing depth [34]. Moreover, it becomes wider with the use of a range shifter and the presence of an air gap, frequently encountered in head and neck cancer patients, to the range of 0.8~1.0 cm in the un-collimated scanning beam condition [35]. Under a similar condition, if a 6 MV photon beam is used, the lateral penumbra width is about 0.3~0.5 cm, measured by an EBT2 film [36]. The patient-specific block collimators for PBT may reduce the penumbra width, which may cause additional heavy workload [37]. The multi-leaf collimator system may be an alternative to mitigate this issue, as shown by Sugiyama et al. [38]. However, collimation systems are not available on the majority of PBT equipment and might lead to additional neutron leakage, which can increase the secondary cancer risk [39]. To compensate for the wider penumbra, an improved collimation technique may serve as a viable example [40,41].

Through the paired rival plan comparisons, we observed that the theoretical merit of the Bragg–Peak phenomenon did not always lead to more favorable dose distribution around the critical neural OARs. We propose that the shortest distance between the target and the critical OARs could serve as a surrogate indicator of choosing the appropriate adaptive re-plan modality, which seems to be a straightforward and practical guidance not only in NPCa patients but also in similar clinical situations.

The current study has a few weak points. This is a retrospective study with rather a small sample size. The follow-up duration was not long enough to report relevant long-term clinical outcomes including RT-related delayed neural toxicities. In addition, NTCP analysis was not attempted. Nevertheless, this study’s main findings could have significant implications in future clinical practice and research in this field. First, the shortest distance between the target and OARs, which was proposed in the current study, could serve as an important and practical criterion in selecting the treatment modality. This could further lead to standardized guideline development, which could further the consistency of treatment plans and improve the overall care quality for patients with cT3-4 NPCa or in similar clinical settings. Second, the appropriate selection of treatment modality, based on our findings, could contribute to the risk reduction of severe side effects, especially neurotoxicity, without compromising the oncologic outcomes. Third, our research could increase the resource utilization efficiency. Based on a clear and reliable selection criterion, unnecessary energy consumption including rival plan generation effort could be avoided.

It is desirable to initiate further studies that could validate our observations, preferably with a prospective study nature with a larger number of patients, in various cancer types. Furthermore, the development of systems could assist radiation oncologists in appropriate and prompt decision-making, which could contribute to more efficient resource utilization.

## 5. Conclusions

Because of the limited IMPT resource, some practical parameters are highly needed to improve clinical outcomes and to increase the resource utilization efficiency. In summary, in order to overcome the above-mentioned innate physical limitations of scanning IMPT, we would propose the shortest distance between the targets and critical OARs as a straightforward and reliable criterion for selecting the RT modality, particularly in the management of cT4 NPCa patients.

## Figures and Tables

**Figure 1 cancers-16-03402-f001:**
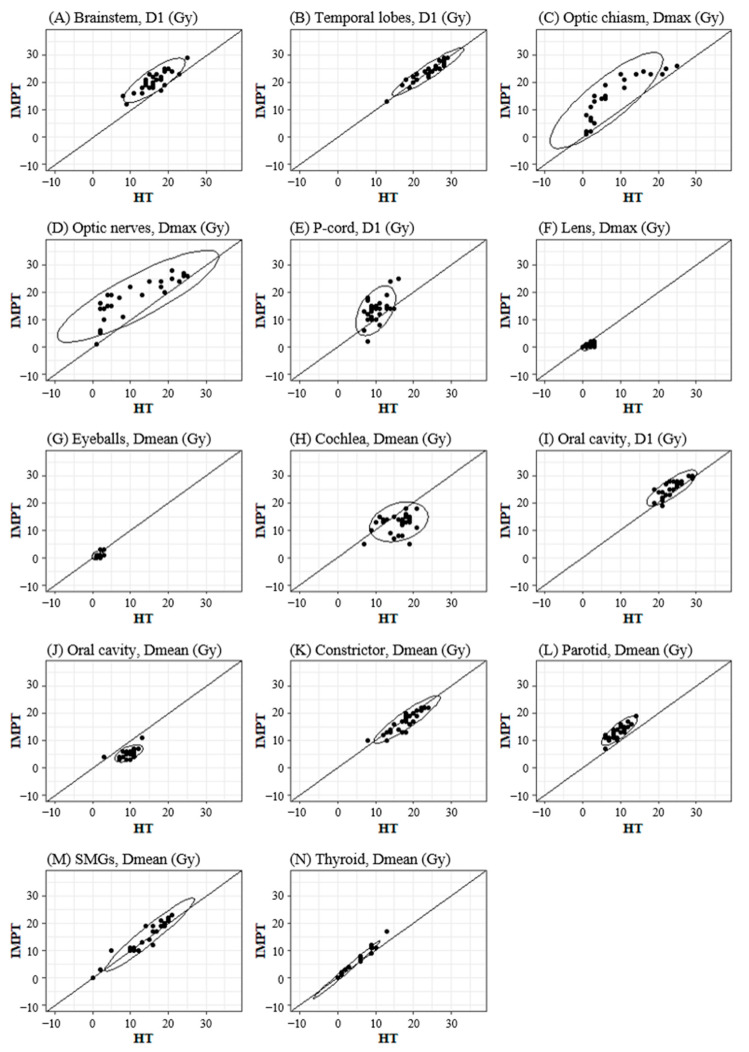
Scatter plots to compare exposed doses to organs at risk (OARs) between the paired helical tomotherapy (HT) and intensity-modulated proton therapy (IMPT) for each patient; ellipse area, 95% confidence interval; solid line, indicating same OAR doses from tomotherapy and proton therapy. Subfigures show comparisons for: (**A**) Brainstem, D1 (Gy); (**B**) Temporal lobes, D1 (Gy); (**C**) Optic chiasm, Dmax (Gy); (**D**) Optic nerves, Dmax (Gy); (**E**) P-cord, D1 (Gy); (**F**) Lens, Dmax (Gy); (**G**) Eyeballs, Dmean (Gy); (**H**) Cochlea, Dmean (Gy); (**I**) Oral cavity, D1 (Gy); (**J**) Oral cavity, Dmean (Gy); (**K**) Constrictor, Dmean (Gy); (**L**) Parotid, Dmean (Gy); (**M**) SMGs, Dmean (Gy); (**N**) Thyroid, Dmean (Gy). D1 represents the dose to 1% of the volume, Dmax is the maximum dose, and Dmean is the mean dose.

**Figure 2 cancers-16-03402-f002:**
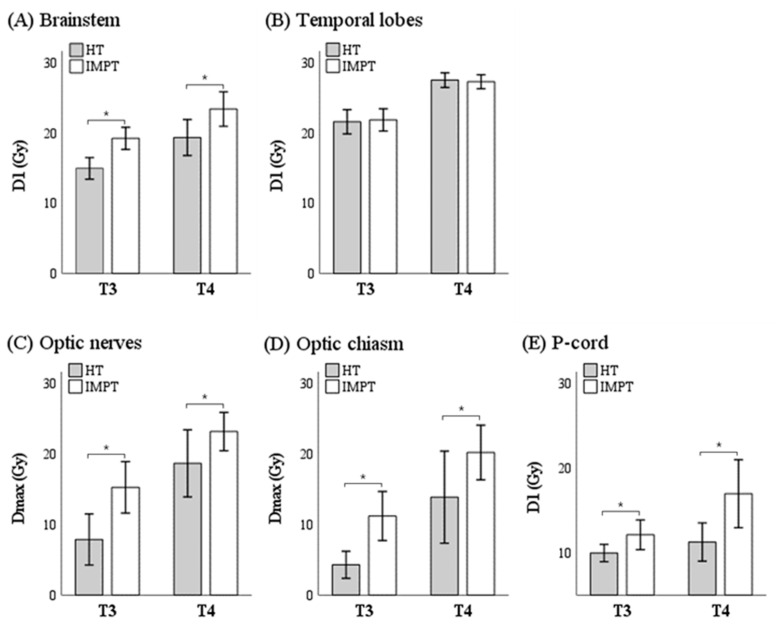
Dosimetric comparison for high-priority organs at risk between the paired plans for helical tomotherapy (HT) and intensity-modulated proton therapy (IMPT) according to tumor stages, T3 and T4. (**A**) Brainstem; (**B**) Temporal lobes; (**C**) Optic nerves; (**D**) Optic chiasm; (**E**) P-cord; box, mean of D1 or Dmax, error bar, 95% confidence interval, * *p* < 0.001 from the paired *t*-test.

**Figure 3 cancers-16-03402-f003:**
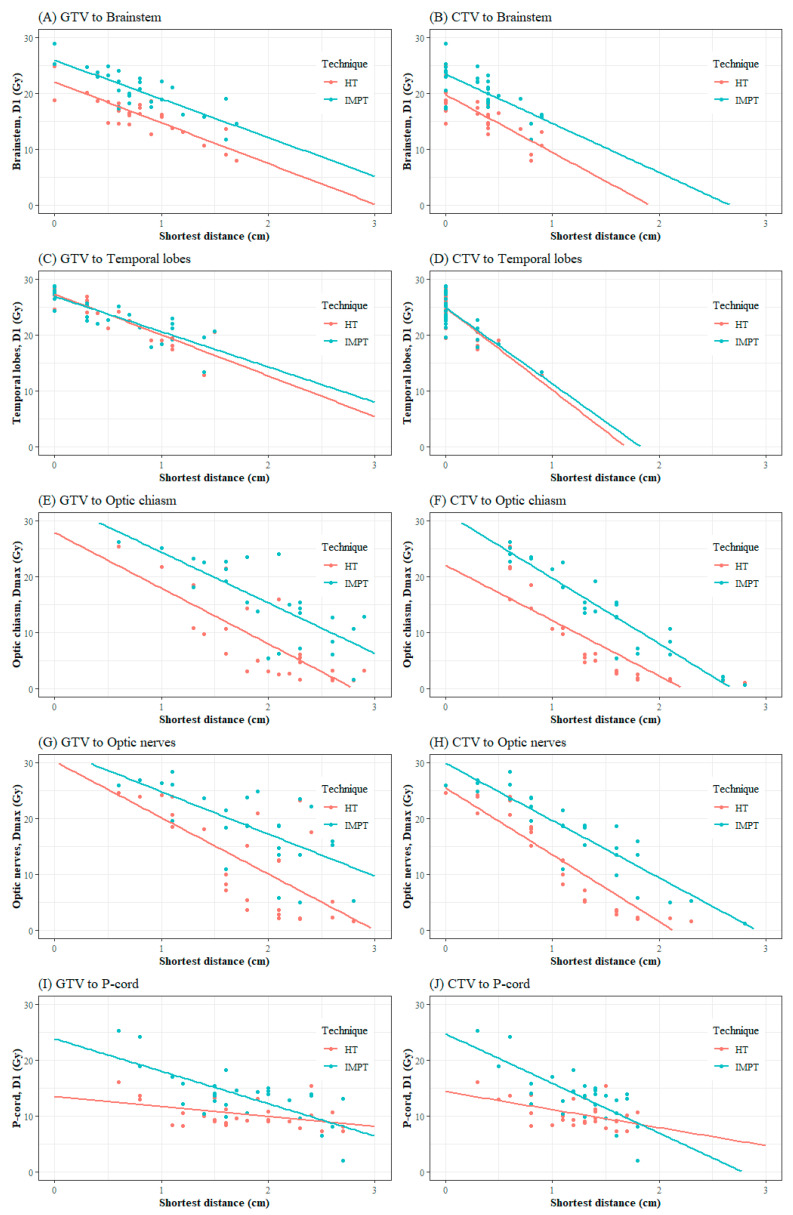
Scatter plots and the linear regression of the shortest distance and doses of OARs in helical tomotherapy (HT) and intensity-modulated proton therapy (IMPT). (**A**,**B**) Brainstem, (**C**,**D**) Temporal lobes, (**E**,**F**) Optic chiasm, (**G**,**H**) Optic nerves, and (**I**,**J**) P-cord, with respect to GTV (left column) and CTV (right column). GTV: Gross Tumor Volume; CTV: Clinical Target Volume.

**Figure 4 cancers-16-03402-f004:**
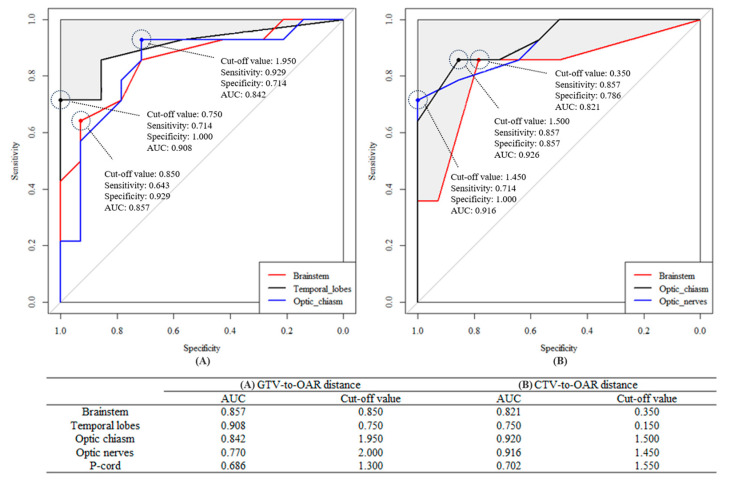
ROC analysis for selection of IMPT with GTV to OAR distance (**A**) and CTV to OAR distance (**B**).

**Table 1 cancers-16-03402-t001:** Patient characteristics (N = 28).

Variables	Total (N = 28)	HT (N = 14)	IMPT (N = 14)	*p*-Value
Median age, years (range)	50 (37–72)	47 (40–65)	50 (37–72)	0.327
Sex				
Female	7 (25.0%)	3 (21.4%)	4 (28.6%)	1
Male	21 (75.0%)	11 (78.6%)	10 (71.4%)	
Smoking status				
Current or Ex-smoker	11 (39.3%)	6 (42.9%)	5 (35.7%)	1
Non-smoker	17 (60.7%)	8 (57.1%)	9 (64.3%)	
Histologic type				
Squamous or keratinizing	4 (14.3%)	2 (14.3%)	2 (14.3%)	1
Non-keratinizing	23 (82.1%)	11 (78.6%)	12 (85.7%)	
Non-specified	1 (3.6%)	1 (7.1%)	-	
cT stage				
cT3	19 (67.9%)	6 (42.9%)	13 (92.9%)	0.013
cT4	9 (32.1%)	8 (57.1%)	1 (7.1%)	
Tumor extent (involved site)				
Intracranial	7 (25.0%)	6 (42.9%)	1 (7.1%)	0.077
Orbital	1 (3.6%)	1 (7.1%)	-	
cN stage				
0–1	9 (32.1%)	6 (42.9%)	11(78.6%)	1
2–3	19 (67.9%)	8 (57.1%)	3 (21.4%)	
Clinical stage				
III	17 (60.7%)	6 (42.9%)	11 (78.6%)	0.122
IV	11 (39.3%)	8 (57.1%)	3 (21.4%)	
Induction chemotherapy				
Not carried out	24 (85.7%)	11 (78.6%)	13 (92.9%)	0.596
Carried out	4 (14.3%)	3 (21.4%)	1 (7.1%)	

**Table 2 cancers-16-03402-t002:** Dosimetric comparison of targets and OAR-related parameters between the paired HT and IMPT plans (N = 28); mean ± standard deviation.

Parameter	HT	IMPT	*p*-Value
Target coverage			
GTV-RF volume (cc)	20.74 ± 17.16		
GTV-RF HI	1.03 ± 0.01	1.04 ± 0.01	<0.001
GTV-RF CI	0.34 ± 0.08	0.40 ± 0.09	<0.001
CTV-RF volume (cc)	177.35 ± 78.47		
CTV-RF HI	1.23 ± 0.01	1.24 ± 0.01	0.073
Dose spillage			
H19:M49CTV-RF CI	0.51 ± 0.04	0.53 ± 0.05	0.204
High, GTV-RF	5.00 ± 2.49	3.87 ± 1.74	<0.001
High, CTV-RF	1.30 ± 0.18	1.39 ± 0.24	0.077
Intermediate, CTV-RF	6.98 ± 0.77	6.63 ± 1.48	0.198
Low, CTV-RF	21.18 ± 2.92	15.96 ± 2.52	<0.001
High-priority OARs			
Brainstem, D1 (Gy)	16.32 ± 3.78	20.52 ± 3.74	<0.001
Temporal lobes, D1 (Gy)	23.43 ± 4.12	23.54 ± 3.78	0.643
Optic chiasm, Dmax (Gy)	7.36 ± 7.24	14.07 ± 7.76	<0.001
Optic nerves, Dmax (Gy)	11.33 ± 8.66	17.77 ± 7.47	<0.001
P-cord, D1 (Gy)	10.37 ± 2.43	13.66 ± 4.70	<0.001
Low-priority OARs			
Lens, Dmax (Gy)	1.39 ± 0.75	0.67 ± 0.59	<0.001
Eyeballs, Dmean (Gy)	1.39 ± 0.65	0.94 ± 0.74	0.001
Cochlea, Dmean (Gy)	15.68 ± 3.55	12.46 ± 3.38	<0.001
Oral cavity, D1 (Gy)	23.57 ± 2.89	25.38 ± 2.90	<0.001
Oral cavity, Dmean (Gy)	9.33 ± 1.92	5.13 ± 1.63	<0.001
Constrictor muscle, Dmean (Gy)	18.01 ± 3.79	17.21 ± 3.81	0.009
Parotid glands, Dmean (Gy)	9.65 ± 2.02	13.30 ± 2.45	<0.001
SMGs, Dmean (Gy)	14.23 ± 5.52	15.01 ± 6.00	0.037
Thyroid, Dmean (Gy)	3.82 ± 4.24	4.38 ± 4.96	0.004
GTV to OAR distance (cm)			
Brainstem	0.5 ± 0.3	1.0 ± 0.4	0.001
Temporal lobes	0.2 ± 0.2	0.9 ± 0.4	<0.001
Optic chiasm	1.7 ± 0.6	2.5 ± 0.6	0.001
Optic nerves	1.6 ± 0.6	2.2 ± 0.7	0.009
P-cord	1.5 ± 0.6	2.0 ± 0.5	0.047
CTV to OAR distance (cm)			
Brainstem	0.2 ± 0.2	0.5 ± 0.3	0.002
Temporal lobes	0	0.2 ± 0.3	0.012
Optic chiasm	1.0 ± 0.4	1.9 ± 0.5	<0.001
Optic nerves	0.7 ± 0.4	1.6 ± 0.5	<0.001
P-cord	1.1 ± 0.4	1.4 ± 0.3	0.06

Abbreviations: HT, helical tomotherapy; IMPT, intensity-modulated proton therapy; GTV-RF, gross tumor volume for reduced field plan; CTV-RF, clinical target volume for reduced field plan; HI, homogeneity index = D2/D98; CI, conformity index = (target volume in the prescription isodose line)2/target volume x volume of the prescription isodose line; OAR, organ at risk; Constrictor m., constrictor muscles; SMGs, submandibular glands.

## Data Availability

The research data are stored in an institutional repository and will be shared upon request to the corresponding authors.

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
