# Peer review of "Dosimetric Comparison and Selection Criteria of Intensity-Modulated Proton Therapy and Intensity-Modulated Radiation Therapy for Adaptive Re-Plan in T3-4 Nasopharynx Cancer Patients"

_cancers, 2024, doi:10.3390/cancers16193402_

Round 1

Reviewer 1 Report

Comments and Suggestions for Authors

The article submitted by Ko et al, titled Dosimetric Comparison & selction Criteria of Intensity-modulated Proton Therapy & Intensity-Modulated Radiation Therapy for Adaptive Re-plan in T3-4 Nasopharynx cancer patients, is well documented piece of study. I have few suggestions,

1) Introduction is well written. Add the locally/globally prevalence of the disease. As this information highlights the need/intervention of new therapeutics.

2) Materials & Methods seems very elaborative. It can be possible shorten.

3)  Add future presepective & potential benefits to the community.

4) Add some new references. As latest reference is 2022.  

Author Response

Reviewer 1:

The article submitted by Ko et al, titled Dosimetric Comparison & selction Criteria of Intensity-modulated Proton Therapy & Intensity-Modulated Radiation Therapy for Adaptive Re-plan in T3-4 Nasopharynx cancer patients, is well documented piece of study. I have few suggestions,

Dear Reviewer,

We sincerely appreciate the time and effort you have invested in reviewing our manuscript. Your insightful comments and constructive suggestions have been invaluable in improving the quality of our work. We have carefully considered all of your feedback and have made substantial revisions to address your concerns. Below, we provide point-by-point responses to each of your comments:

1) Introduction is well written. Add the locally/globally prevalence of the disease. As this information highlights the need/intervention of new therapeutics.

We added the following 2 sentences in the introduction and modified a few sentences accordingly.

“Nasopharynx cancer (NPCa) is rather rare disease, with the global age-adjusted incidence rate of 1.5 per 100,000 person-year in 2020 [1]. The incidence rate of NPCa has been reported the highest in Eastern Asia (including China) and South-Eastern Asia.”

2) Materials & Methods seems very elaborative. It can be possible shorten.

We have revised the Materials & Methods section to be more concise. The key information has been retained while reducing unnecessary details and overall length.

3) Add future perspective & potential benefits to the community.

We added the following sentences in the late part of Discussion.

“The current study has a few weak points. This is a retrospective study with rather a small sample size. The follow-up duration was not long enough to report the relevant long-term clinical outcomes including RT-related delayed neural toxicities. In addition, NTCP analysis was not tried. Nevertheless, this study's main findings could have significant implications in future clinical practice and research in this field. First, the shortest distance between the target and OARs, which was proposed in the current study, could serve as an important and practical criterion in selecting the treatment modality. This could further lead to the standardized guideline development which could endorse the consistency and improve the overall care quality for the patients with cT3-4 NPCa or with similar clinical settings. Second, the appropriate selection of treatment modality, based on our findings, could contribute to the risk reduction of severe side effects, especially neuro-toxicity, without compromising the oncologic outcomes. Third, our research could increase the resource utilization efficiency. Based on a clear and reliable selection criterion, unnecessary energy consumption including rival plan generation effort could be avoided.

It is desirable to undergo further studies that could validate our observations, prefer-ably in prospective study nature with larger patients’ number, in various cancer types. Furthermore, the development of systems that could assist the radiation oncologists in ap-propriate and prompt decision-making, which could contribute to more efficient resource utilization.”

4) Add some new references. As latest reference is 2022.

We have added new recent references to our manuscript. Specifically, we have included two papers published in 2023 as references [23, 24].

We believe that these revisions have significantly strengthened our manuscript and hope that they adequately address your concerns. Once again, we thank you for your valuable input and the opportunity to improve our work.

Reviewer 2 Report

Comments and Suggestions for Authors

Congratulation for Authors!

This study is extremely nice and appropriate! I wish I could do that...

My only question is, in case of induction therapy did you detect downstaging?

I highly support the manuscript to accept.

Author Response

Reviewer 2:

This study is extremely nice and appropriate! I wish I could do that...

Dear Reviewer,

We sincerely appreciate the time and effort you have invested in reviewing our manuscript.

Below, we provide responses to your comment:

My only question is, in case of induction therapy did you detect downstaging?

We feel sorry as we cannot answer this question specifically. In general, response evaluation following induction chemotherapy in cT3-4 cases is usually very difficult. And though tumor burdens might have been reduced following induction chemotherapy, the target volume reduction on the primary tumor usually is not feasible, especially if the skull base is involved. In addition, it is typically recommended that the post-induction target should be delineated based on the pre-induction gross tumor volume, which usually negates the effect of induction chemotherapy in cT3-4 cases.

I highly support the manuscript to accept.

Reviewer 3 Report

Comments and Suggestions for Authors

In their manuscript "Dosimetric Comparison and Selection Criteria of Intensity Modulated Proton Therapy and Intensity-Modulated Radiation  Therapy for Adaptive Re-plan in T3-4 Nasopharynx Cancer Patients" ,the authors presents a comprehensive compare result that NPCa patients at cT4 stage or with the shortest distance between target and critical neural structures <0.8 cm were suboptimal candidates for IMPT adaptive re-planning. The manuscript can hardly be evaluated in its present form as the supplementry data did not include in study . Listing following issues are some concerns:

1 English editing is required. This should be clearly stated in the reply letter and the person should be included in the acknowledgments.

2 The sample size 28 is less enough to verify the current conclusion ,how did you overcome the data bias?

3 In result ,you indicated that "Receiver operating characteristic curves analysis suggested a 0.8 cm cut-off between gross tumor volume and brainstem/temporal lobes for selecting IMPT  (specificity=1) .please explain how did specificity=1 achieve?in addition ,ROC data showing in main text is better to reader for understanding.

4 the Table 1 and 3 may be the same content to covey patients base line ,why did you show Table  1?

5 The dose relaying side effect is also important ,it is better to compare their toxicity 

6 Supplementary Materials are not included in study.

7 it is better to discusse what the trend of decision-making of the treatment modality Adaptive Re-plan in T3-4 Nasopharynx Cancer Patients in future?

Comments on the Quality of English Language

need language edit

Author Response

Reviewer 3:

In their manuscript "Dosimetric Comparison and Selection Criteria of Intensity Modulated Proton Therapy and Intensity-Modulated Radiation Therapy for Adaptive Re-plan in T3-4 Nasopharynx Cancer Patients", the authors present a comprehensive compare result that NPCa patients at cT4 stage or with the shortest distance between target and critical neural structures <0.8 cm were suboptimal candidates for IMPT adaptive re-planning.

Dear Reviewer,

We sincerely appreciate the time and effort you have invested in reviewing our manuscript. Your insightful comments and constructive suggestions have been invaluable in improving the quality of our work. We have carefully considered all of your feedback and have made substantial revisions to address your concerns. Below, we provide point-by-point responses to each of your comments:

The manuscript can hardly be evaluated in its present form as the supplementary data did not include in study. Listing following issues are some concerns:

1 English editing is required. This should be clearly stated in the reply letter and the person should be included in the acknowledgments.

English revision by native speaker, who does not major in medical science, is frequently inappropriate and sometimes twist the objective findings. We intended to avoid this distortion, and tried our best to modify many phrases and sentences to improve the English quality.

2 The sample size 28 is less enough to verify the current conclusion, how did you overcome the data bias?

We agree with the reviewer’s opinion that data on more enough sample size would be more powerful. Overall, the number of newly diagnosed NPCa patients per year in Korea is around 400+. Considering this, recruiting 28 patients with cT3-4 disease during 16 months’ period is not easy in Korea, and this could have been possible since our institute has accommodated about 15%~20% of these patients’ load annually (most probably the largest load in Korea). We were very keen to find out the appropriate indicator assisting the treatment modality selection as early as possible, and could find the reasonable parameter of “the shortest distance” between the target and critical OARs based on 28 patients. Recruiting more patients would be desirable, however, it would not only take longer time but also require more energy consumption in generating the rival plans, which seems not easy considering the relative resource limitation at the authors’ institute.

3 In result, you indicated that "Receiver operating characteristic curves analysis suggested a 0.8 cm cut-off between gross tumor volume and brainstem/temporal lobes for selecting IMPT (specificity=1). please explain how did specificity=1 achieve? in addition, ROC data showing in main text is better to reader for understanding.

This was a typo. We revised the abstract and the results including Figure 4.

4 the Table 1 and 3 may be the same content to covey patients’ baseline, why did you show Table 1?

The original Tables 1 and 3 contained overlapping information. We have eliminated Table 3 and incorporated the necessary contents into modified Tables 1 and 2.

5 The dose relaying side effect is also important, it is better to compare their toxicity 

We agree with the reviewer’s opinion. The current study mainly focused on the neural tissue damage risk in cT3-4 NPCa patients by 2 different adaptive RT modalities. Because RT-related neural toxicities usually appear several years later, it is difficult to make specific comments on them in this study. Instead, we mentioned this point in the Discussion as a limitation of our study.

6 Supplementary Materials are not included in study.

We will upload the supplement data.

7 it is better to discuss what the trend of decision-making of the treatment modality Adaptive Re-plan in T3-4 Nasopharynx Cancer Patients in future?

There has been few evidences regarding the treatment modality selection in adaptive re-plan. We suggested a practical guidance and added the following sentences in the discussion.

“The current study has a few weak points. This is a retrospective study with rather a small sample size. The follow-up duration was not long enough to report the relevant long-term clinical outcomes including RT-related delayed neural toxicities. In addition, NTCP analysis was not tried. Nevertheless, this study's main findings could have significant implications in future clinical practice and research in this field. First, the shortest distance between the target and OARs, which was proposed in the current study, could serve as an important and practical criterion in selecting the treatment modality. This could further lead to the standardized guideline development which could endorse the consistency and improve the overall care quality for the patients with cT3-4 NPCa or with similar clinical settings. Second, the appropriate selection of treatment modality, based on our findings, could contribute to the risk reduction of severe side effects, especially neuro-toxicity, without compromising the oncologic outcomes. Third, our research could increase the resource utilization efficiency. Based on a clear and reliable selection criterion, unnecessary energy consumption including rival plan generation effort could be avoided.

It is desirable to undergo further studies that could validate our observations, prefer-ably in prospective study nature with larger patients’ number, in various cancer types. Furthermore, the development of systems that could assist the radiation oncologists in ap-propriate and prompt decision-making, which could contribute to more efficient resource utilization.”

We believe that the above revisions, which were made according to the reviewers’ comments, have significantly strengthened our manuscript, and would hope that they adequately address your concerns. Once again, we thank you for your valuable input and the opportunity to improve our work.

Round 2

Reviewer 3 Report

Comments and Suggestions for Authors

Accept ,but you should explain the question reasonablely  "In result, you indicated that "Receiver operating characteristic curves analysis suggested a 0.8 cm cut-off between gross tumor volume and brainstem/temporal lobes for selecting IMPT (specificity=1). please explain how did specificity=1 achieve? in addition, ROC data showing in main text is better to reader for understanding." it was explained in typo, but  In abstract ,there is no ROC data.please clarify it . 

Author Response

Accept ,but you should explain the question reasonablely  "In result, you indicated that "Receiver operating characteristic curves analysis suggested a 0.8 cm cut-off between gross tumor volume and brainstem/temporal lobes for selecting IMPT (specificity=1). please explain how did specificity=1 achieve? in addition, ROC data showing in main text is better to reader for understanding." it was explained in typo, but  In abstract ,there is no ROC data.please clarify it . 

Dear Reviewer,

Thank you for your valuable feedback on our manuscript. We appreciate your suggestion to include more detailed ROC data in the abstract for better clarity and understanding.

In response to your comment, we have added the following sentence to the Results section of the abstract:

"Receiver operating characteristic curve analyses were done to find out the optimal cut-off values of the shortest distances between the target and the OARs (temporal lobes and brainstem), which were 0.75 cm (AUC=0.908, specificity=1.00) and 0.85 cm (AUC=0.857, specificity=0.929), respectively;"

We have also ensured that this information is consistent with the more detailed methods and results presented in the main text and Figure 4. We hope that this revision addresses your concern and improves the overall clarity of our abstract.

Thank you again for your insightful comments.